# Naturalistic Hyperscanning with Wearable Magnetoencephalography

**DOI:** 10.3390/s23125454

**Published:** 2023-06-09

**Authors:** Niall Holmes, Molly Rea, Ryan M. Hill, Elena Boto, James Leggett, Lucy J. Edwards, Natalie Rhodes, Vishal Shah, James Osborne, T. Mark Fromhold, Paul Glover, P. Read Montague, Matthew J. Brookes, Richard Bowtell

**Affiliations:** 1Sir Peter Mansfield Imaging Centre, School of Physics and Astronomy, University of Nottingham, University Park, Nottingham NG7 2RD, UK; ryan.hill@nottingham.ac.uk (R.M.H.); james.leggett@nottingham.ac.uk (J.L.); lucy.edwards@nottingham.ac.uk (L.J.E.); natalie.rhodes@nottingham.ac.uk (N.R.); paul.glover@nottingham.ac.uk (P.G.); matthew.brookes@nottingham.ac.uk (M.J.B.); richard.bowtell@nottingham.ac.uk (R.B.); 2Cerca Magnetics Limited, Unit 2 Castlebridge Office Village, Kirtley Drive, Nottingham NG7 1LD, UK; molly@cercamagnetics.com (M.R.); elena@cercamagnetics.com (E.B.); 3QuSpin Inc., 331 South 104th Street, Suite 130, Louisville, CO 80027, USA; vshah@quspin.com (V.S.); josborne@quspin.com (J.O.); 4School of Physics and Astronomy, University of Nottingham, University Park, Nottingham NG7 2RD, UK; mark.fromhold@nottingham.ac.uk; 5Fralin Biomedical Research Institute, Department of Physics, Virginia Tech, Roanoke, VA 24016, USA; read@vtc.vt.edu

**Keywords:** optically pumped magnetometer, magnetoencephalography, hyperscanning, electromagnetic coil, on-scalp MEG, magnetic shielding

## Abstract

The evolution of human cognitive function is reliant on complex social interactions which form the behavioural foundation of who we are. These social capacities are subject to dramatic change in disease and injury; yet their supporting neural substrates remain poorly understood. Hyperscanning employs functional neuroimaging to simultaneously assess brain activity in two individuals and offers the best means to understand the neural basis of social interaction. However, present technologies are limited, either by poor performance (low spatial/temporal precision) or an unnatural scanning environment (claustrophobic scanners, with interactions via video). Here, we describe hyperscanning using wearable magnetoencephalography (MEG) based on optically pumped magnetometers (OPMs). We demonstrate our approach by simultaneously measuring brain activity in two subjects undertaking two separate tasks—an interactive touching task and a ball game. Despite large and unpredictable subject motion, sensorimotor brain activity was delineated clearly, and the correlation of the envelope of neuronal oscillations between the two subjects was demonstrated. Our results show that unlike existing modalities, OPM-MEG combines high-fidelity data acquisition and a naturalistic setting and thus presents significant potential to investigate neural correlates of social interaction.

## 1. Introduction

Human social interaction is at the core of healthy neurodevelopment. From tactile stimulation to the evolution of language, from information transfer to social development, how we interact with others shapes everything from our abilities and skills to our personalities. However, relatively little is known about the neural underpinnings of these interactions. The simultaneous recording of functional brain imaging data from multiple people (hyperscanning) offers a powerful tool to probe brain activity underlying social interaction [1,2]. However, the available functional imaging technology places limitations on experimental design, participant experience and data quality [3].

Functional magnetic resonance imaging (fMRI) offers an assessment of brain activity with high spatial resolution, but the requirement that participants be enclosed and motionless in a noisy scanner makes natural interactions during hyperscanning impossible. Whilst MRI can be adapted to scan two people simultaneously [4,5], this results in an environment which offers limited possibilities for experimental design. Most fMRI hyperscanning studies [6,7] have used separate scanners connected via video, but this imposes further barriers to natural interaction. In addition, the latency and longevity of the haemodynamic signal make it challenging to use in assessing brain dynamics.

In contrast, functional near-infrared spectroscopy (fNIRS) [8] and electroencephalography (EEG) [9] are wearable technologies that can be deployed in real-life settings [10,11,12], enabling more naturalistic hyperscanning. However, fNIRS suffers poor spatial resolution and (like fMRI) is limited to haemodynamic measurement. EEG, via assessment of scalp-level electrical potentials, directly measures brain electrophysiology and consequently has excellent temporal resolution but suffers from poor spatial resolution and is sensitive to artefacts from non-neuronal sources of electrical activity, especially muscles, during participant movement [13].

Magnetoencephalography (MEG) measures the magnetic fields generated by neuronal currents [14], providing a direct assessment of electrophysiology. Unlike EEG, MEG has a high spatial precision [15,16] and lower sensitivity to non-neuronal artefacts [17]. However, conventional MEG systems use cryogenically cooled superconducting quantum interference devices (SQUIDs) [18] housed in magnetically shielded rooms (MSRs) to gain sufficient sensitivity to measure the neuromagnetic fields. Low-temperature operation means that sensors must be positioned in a fixed array, 2–3 cm from the scalp (to provide thermal insulation). So, like MRI scanners, conventional MEG systems are cumbersome and static; only one person can be scanned at once, participants must remain motionless and performance is limited by the significant scalp-sensor separation. Nevertheless, the potential for hyperscanning has been demonstrated using two MEG systems sited in the same MSR [19], and geographically displaced systems connected via video [20,21]. Sequential dual-brain imaging studies have also been performed [22,23] with participants viewing videos of social interaction.

In summary, hyperscanning experiments can be carried out with existing brain imaging technology, and such studies are beginning to provide unique insights into how the human brain mediates social interaction [3]. However, current instrumentation is limited either by its performance (EEG/fNIRS) or the unnatural scanning environment it provides (MEG/fMRI).

Recently, ‘wearable’ MEG has been developed through the use of optically pumped magnetometers (OPMs) [24], see [25] for a review. OPMs are sensitive magnetic field sensors that do not require cryogenics. These devices have enabled the design of flexible MEG sensor arrays, which can be placed closer to the scalp, and adapted to the requirements of individual studies and participants. Increased proximity to the scalp can improve sensitivity and spatial resolution beyond that which is achieved using a cryogenic MEG [26,27]. In addition, the lightweight nature of OPMs has enabled the development of wearable systems which allow participants to move during recordings [24,28,29,30] and enable scanning across the lifespan [31,32]. The motion tolerance and flexibility of the application, coupled with the provision of high-fidelity data, mean that OPM-MEG forms an ideal platform for hyperscanning.

However, to achieve sensitivity to the MEG signals, OPM-MEG requires a strict zero magnetic field environment [33]. Furthermore, OPMs are vector magnetometers meaning any movement of a sensor through a non-zero background field will generate artefacts that can mask brain activity and saturate sensor outputs. These constraints mean that OPM-MEG experiments involving participant motion require an OPM-optimised MSR with a very low remnant magnetic field along with ‘active’ magnetic shielding in the form of electromagnetic coils which cancel both the remnant magnetic field and field changes experienced by the array [34,35,36,37,38]. Using such shielding systems to attenuate the field at the OPM array has been shown to allow naturalistic motion during MEG studies of an individual subject [24,34] demonstrating the power of OPM-MEG as a neuroscientific tool. Hyperscanning using OPM-MEG introduces the additional challenge of requiring simultaneous field cancellation at the OPM arrays worn by two subjects.

Here, we describe the first OPM-MEG hyperscanning experiments which were carried out using an array of OPMs divided across participants and a matrix coil system [39] for field cancellation. In what follows, we show results from two recordings: first, a two-person touching task to induce controlled motor and sensory responses and second, a two-person ball game where participants bat a table-tennis ball back and forth. These studies show the potential of naturalistic OPM-MEG hyperscanning.

## 2. Materials and Methods

### 2.1. OPM-MEG System

The OPM-MEG system used here (except the matrix coils) is described in detail by Hill et al. [40]; here, we outline briefly its main features. The system (shown schematically in Figure 1) is housed inside an MSR which is optimised for OPM operation (MuRoom, Cerca Magnetics Limited, Nottingham, UK). The MSR features 4 layers of mu-metal and 1 layer of copper, along with demagnetisation coils [41,42]. The typical remnant magnetic fields and field gradients at the centre of the room are of order 2 nT and 2 nT/m, respectively.

At the time of recording, up to 50, second generation, QuSpin Inc. (Louisville, CO, USA) zero-field magnetometers were available for array formation (see Tierney et al. [43] and Schofield et al. [44] for reviews of OPM physics and Osborne et al. [45] for specific details of the QuSpin sensor). The OPMs were mounted inside 3D-printed, rigid helmets which allow co-registration of OPM positions and orientations to anatomical MRIs of the participants’ heads [40,46,47] (whole-head MRI scans were generated using a 3 T Philips Ingenia system, running an MPRAGE sequence, at an isotropic spatial resolution of 1 mm). OPMs were configured to record only the component of the magnetic field which is radial to the surface of the head. OPM data were sampled at 1200 Hz using a series of National Instruments (NI, Austin, TX, USA) NI-9205 16-bit analogue to digital converters interfaced with LabVIEW (NI, Austin, TX, USA). Since all the OPMs are sampled and controlled using the same equipment, no additional timing signals or hardware is required to synchronise the data collected from the two helmets.

Participant movements were tracked using an OptiTrack V120:Duo (NaturalPoint Inc., Corvallis, OR, USA) optical tracking system which allows accurate optical tracking of multiple rigid bodies at a sample rate of 120  Hz. Two cameras, each with an array of 15 infrared (IR) LEDs, are used to illuminate IR reflective markers and the combined coordinates of multiple markers are used to form a rigid body tracking with 6 degrees of freedom (x, y and z translations, pitch, yaw and roll rotations).

### 2.2. Matrix Coils

Our aim was to develop an electromagnetic coil system that produces a magnetic field, equal in magnitude but opposite in direction to the remnant magnetic field within two target volumes inside the MSR, thereby nulling the field and enabling the required zero-field environment for OPM operation across two helmets. Matrix coil systems feature an array of small, simple, unit coils positioned around the participant. Superposition of the magnetic field generated by multiple coils, each carrying an independently controllable current, enables the production of a wide range of patterns of magnetic field variation within a selected target volume [48,49]. Similar multi-coil shimming systems have been developed for MRI [50,51]. Our matrix coil system was constructed using a bi-planar design, with each plane containing 24 square coils (square side length 38 cm). The coils are arranged on a 4 × 4 grid with an overlapping 3 × 3 grid in which the central coil is omitted (Figure 1. Each coil was wound by hand using 10 turns of 0.56 mm diameter copper wire, tightly wrapped around a series of plastic guides attached to a wooden structure (coil resistance ~2 Ω, coil inductance ~160 µH). The two planes are sited on either side of the participant(s), separated by 150 cm.

Each unit coil is connected to a single output of a 48-channel, low-noise, voltage amplifier that was designed and constructed in-house. The amplifiers are interfaced with three NI-9264 16-bit, digital-to-analogue converters (DACs) that are controlled using LabVIEW. The voltages applied at the amplifier input range between ±10 V (least significant bit (lsb) voltage = 20 V/2^16^ = 0.305 mV). The maximum electrical current in the coil is tuned by an additional series resistance, which here was 1.2 kΩ. This was chosen to control the magnetic field noise generated by the coils. Specifically, the coil driver current noise at this resistance is <10 nA/√Hz in the 1–100 Hz band, we estimate this translates to <20 fT/√Hz noise in the field from all 48 coils at the centre of the planes. For comparison, the OPM noise floor is <15 fT/√Hz in this frequency range so the two are comparable. The maximum current which can be applied to each coil is ±8.33 mA, and the lsb current is 2.54 µA.

The matrix coil has previously been used to enable field nulling of a single moving helmet by combining OPM measurements with optical tracking and simulation of fields generated by the unit coils [39]; here, we employed a different, data-driven, approach to enable hyperscanning. If the magnetic field measured by the n^th^ OPM in an array of N sensors spread across two helmets due to a unit current in the m^th^ coil in a set of M (=48) matrix coils is written as dbndIm, we can form a (N × M) coil calibration matrix, A, from the full set of values. The field nulling problem can then be described using the following matrix equation:(1)db1dI1db1dI2…db1dIMdb2dI1db2dI2…db2dIM⋮⋮⋱⋮dbNdI1dbNdI2…dbNdIMI1I2⋮IM=−b1b2⋮bN,
(2)Ax=−b.
where the (M × 1) column vector x contains the currents applied to each coil and the (N × 1) column vector b characterises the magnetic field to be cancelled. b is formed using the DC field values measured at the sensors, the negative sign is used to ensure the calculated currents null the magnetic field measured by the array.

The coil currents required to minimise the sum of squares of the measured magnetic field values can be found by identifying the Moore–Penrose pseudo-inverse matrix of A,
(3)$$x=-{\left(AA^{T}\right)}^{-1}A^{T}b.$$

To minimise the power dissipated by the system, and ensure the solution is physically realisable, the matrix $AA^{T}$ can be regularised prior to inversion by addition of a matrix αI, where I is the identity matrix of the same dimensions as $AA^{T}$ and α is a regularisation parameter, i.e.,
(4)$$x=-{\left(AA^{T}+\alpha I\right)}^{-1}A^{T}b.$$

To keep the coil currents within the allowed bounds, Equation (4) is cast as a feed-forward controller: Coil currents are incrementally updated, based on the OPM field measurements at each timepoint i, and the currents applied at the preceding time point as
(5)$$x^{i}=x^{i-1}-G{\left(AA^{T}+\alpha I\right)}^{-1}A^{T}b^{i}.$$

The gain coefficient G and regularisation parameter α are set empirically by monitoring the values of the applied coil currents and the field vector b to produce a stable reduction in the measured fields towards zero on a timescale of a few seconds. If G is too small, the system will take a long time to reach a null. Conversely, if G is too large, the system will become unstable and the field values may begin to oscillate uncontrollably. If α is too small, the fields produced will be specifically tuned to the sensor positions, meaning any error in the measurement (of the coil calibration matrix or the residual field) will be amplified, large coil currents will be required (which may be impossible to generate using the system electronics) and the field variation between sensor positions may not be spatially smooth. If α is too large, the field generated will be unable to reproduce the spatial field variations required for field nulling.

This approach can readily be adapted to changes in the number and shape of the unit coils and flexibly incorporates multiple sensor arrays. However, it only considers the field values at the sensor positions, and as a result, unwanted deviations in the magnetic field could occur between target points. Coil calibration data for populating the matrix, A, can be collected in a variety of ways depending on the available sensing technology, e.g., by pulsing each coil in turn or by applying a known sinusoidal current to each coil. Values can also be calculated based on known sensor positions, coil design and geometry of the MSR [39].

The nulling procedure described above was implemented in LabVIEW. Participants were instructed to remain still whilst a 5 V (4.16 mA), 100 ms pulse was applied to each coil in turn. The change in the field experienced by each OPM was measured by interfacing the OPMs with LabVIEW and operating the sensors in their ‘field-zeroing’ mode. In this mode, QuSpin OPMs can measure the DC magnetic field experienced by the cell (along two orthogonal directions) with a dynamic range of ±50 nT [45,52]. The field zeroing procedure is generally carried out prior to OPM gain calibration when an experiment is performed, providing measurements of the offset magnetic fields required to produce the zero-field environment in the sensor cell. The regularisation parameter α (in Equation (5)) was set to 1% of the maximum singular value of the matrix $AA^{T}$. The feed-forward controller gain was set to 0.1 with a time step of 100 ms.

The time needed for the calibration process scales with the number of coils and takes around 1 min to complete for the 48-coil system. The final coil currents were held constant during the experiments, i.e., no dynamic tracking of changes in the magnetic field was applied (we note that the magnetic field drift in our MSR is on the order of 200 pT over 10 min). The LabVIEW program stores the magnetic field values reported by each sensor prior to calibration, along with the coil calibration matrix, the final voltages applied to each coil and the final magnetic field values.

### 2.3. Data Acquisition

All data were collected by the authors. Participants provided written informed consent for all experiments. All studies were approved by the University of Nottingham’s Faculty of Medicine and Health Sciences Research Ethics Committee. Additional guidelines to mitigate the risk of transmission of COVID-19 were adhered to by all participants and experimenters: Participants wore face masks and visors during the two-person experiments.

## 3. Two-Person Touching Task

### 3.1. Methods

We first explored the capabilities of OPM-MEG hyperscanning by conducting a simple, guided, two-person touch experiment.

Each participant wore an array of 16 OPMs placed over the left sensorimotor cortex mounted in a 3D-printed helmet. (Figure 2a,b shows sensor positions and orientations with respect to each participant’s head.) Participant 1 (female, aged 30, height 172 cm) wore a helmet which was custom-made for their head based on an anatomical MRI (Chalk Studios, London, UK) meaning that co-registration of the positions and orientations of the OPMs with respect to the participant’s brain was known [46]. Participant 2 (male, aged 25, height 182 cm) wore a rigid, additively manufactured generic scanner-cast (Added Scientific Limited, Nottingham, UK) which was designed to fit an average adult head shape [40]. The co-registration of OPM sensors to the anatomy of Participant 2 was performed by using 3D-structured light scans combined with the known structure of the generic helmet [40,47].

Field nulling was performed using measurements of the amplitude of two field components from 10 OPMs operating in field zeroing mode housed in the scanner-casts of each participant (i.e., the matrix A contains values from 20 OPMs giving N=40 measurements in total) as inputs to the LabVIEW-based field nulling program described above. The 10 sensors on each helmet that were used for the nulling process are shown in Figure 2, these included sensors sited at the front, back and right-hand sides of the head (to extend the region of space over which fields were considered in the nulling process), as well as seven sensors sited over the left side of the head. Participants were asked to remain still whilst the system was calibrated and instructed to keep their feet planted throughout the experiment to avoid translating their heads away from the nulled field volume. Figure 3a shows the reduction in amplitude of the field strength over the helmets before and after the nulling is applied. The mean and standard deviation across sensors of the absolute values of the remnant field reported by the nulling sensors decreased from 3.3 ± 1.8 nT to 0.48 ± 0.44 nT (a factor of 6.9) for Participant 1 and from 4.1 ± 2.2 nT to 0.43 ± 0.40 nT (a factor of 9.5) for Participant 2.

All OPM data were collected at a sample rate of 1200 Hz using the equipment described earlier. Once the matrix coil currents had been set, the OPMs were field zeroed and calibrated using the QuSpin software (V7.6.2). The OPMs were then set to their 0.33× gain mode (voltage to magnetic field conversion factor 0.9 V/nT) in which their dynamic range is ±5 nT.

The experimental setup is shown in Figure 4a. Two participants stood on either side of a table, ~65 cm apart. Upon hearing an audio cue, Participant 1 (female, right-handed, age 30, height 172 cm) reached over the table with their right hand and stroked the back of the right hand of Participant 2 (male, right-handed, age 25, height 182 cm). Upon a second (different) audio cue, the roles were reversed. Trials were defined as either ‘odd’ (Participant 1 touches Participant 2, Participant 1 is the ‘active’ Participant) or ‘even’ (Participant 2 touches Participant 1, Participant 2 active). The sequence was repeated 30 times (60 trials total) in the same order with an inter-trial interval of 5 s. The movements of the two OPM-MEG helmets were recorded using an optical tracking system.

We hypothesised that beta band (13–30 Hz) modulation, as a result of the motor control or sensory response, would be observable in the primary sensorimotor regions [53]. To test this, we used a linearly constrained minimum variance beamformer [54] to derive images of oscillatory modulation during the task. Briefly, an estimate of the neuronal current dipole strength, ${\hat{Q}}_{\theta }(t)$, at time t and a position and orientation θ in the brain is formed via a weighted sum of the measured data as
(6)$${\hat{Q}}_{\theta }\left(t\right)=w_{\theta }^{T}m\left(t\right)$$
where m(t) is a vector containing the magnetic field measurements recorded by all OPMs and $w_{\theta }$ is a weights vector tuned to θ. The weights are chosen such that
(7)$$\min\left[{\hat{Q}}_{\theta }^{2}\right]s.t.w_{\theta }^{T}L_{\theta }=1$$
where $L_{\theta }$ is the forward field vector containing the solutions to the MEG forward problem for a unit dipole at θ. The optimal weights vector is expressed as
(8)$$w_{\theta }^{T}={\left[L_{\theta }^{T}{\left\{C+\mu I\right\}}^{-1}L_{\theta }\right]}^{-1}L_{\theta }^{T}{\left\{C+\mu I\right\}}^{-1}$$
where C is the sensor data covariance matrix. Inversion of the covariance matrix is aided by Tikhonov regularisation (i.e., by the addition of the identity matrix scaled by regularisation parameter μ).

To compute the weights vectors for each experiment, the entire dataset was filtered to the beta band (13–30 Hz) and used to compute the covariance matrix. The regularisation parameter μ was set to 0.01 times the largest singular value of the unregularised covariance matrix. The forward field vector was calculated using a multi-sphere head model and the current dipole approximation [55].

Images of activation show the pseudo-T-statistical contrast between data recorded in active and control windows. Specifically, two covariance matrices were computed for the active and control periods, $C_{a}$ and $C_{c}$, respectively, and the pseudo-T-statistical contrast, at θ, was calculated as
(9)$${Ŧ}_{\theta }=\frac{w_{\theta }^{T}{C_{a}w}_{\theta }-w_{\theta }^{T}C_{c}w_{\theta }}{2w_{\theta }^{T}C_{c}w_{\theta }}.$$

Pseudo-T-statistics were computed at the vertices of a regular 4 mm grid spanning the whole brain. This grid of Ŧ values was then thresholded to a percentage of the maximum value and overlaid onto the anatomical MRI of each participant.

We also performed a time–frequency analysis to show the modulation of neural oscillations at the voxel with the largest pseudo-T-statistic. The signal from this peak location was reconstructed to form a ‘virtual electrode’ timecourse (using Equation (6)), with beamformer weights calculated in the broadband (1–150 Hz). A time–frequency spectra were then generated by filtering this timecourse sequentially into overlapping frequency bands between 1 and 50 Hz (1–4 Hz, 2–6 Hz, 4–8 Hz, 6–10 Hz, 8–13 Hz, 10–20 Hz, 15–25 Hz, 20–30 Hz, 25–25 Hz, 30–40 Hz, 35–45 Hz, 40–50 Hz and 45–55 Hz). For each band, the Hilbert envelope was calculated before segmenting and averaging over trials and concatenating in the frequency domain. The mean envelope in the beta band was computed using the virtual electrode timecourse filtered to the beta band (13–30 Hz).

### 3.2. Results

Figure 4b shows beamformer images contrasting the task (0.5 s < t < 2 s) and control (3 s < t < 4.5 s) time windows, the Ŧ values were thresholded to 80% of the maximum value for each condition. The spatial pattern of activation suggests activity in the sensorimotor regions.

Figure 4c–e show the temporal dynamics of oscillatory power at the peak voxel location. Despite the large head movements which participants made as they reached across the table (maximum translations from the starting position in any one trial were 16 mm and 24 mm for Participants 1 and 2, respectively, and the maximum rotations were 3.0° and 7.9°), the task induced a reduction in beta amplitude. In each case, the active participant (i.e., the one performing the touch) showed a reduction in beta power that commenced earlier and persisted longer than that seen in the passive participant. This experiment demonstrates that high-quality OPM-MEG hyperscanning data can be obtained using our system, even in the presence of movements.

## 4. Two-Person Ball Game

### 4.1. Methods

To further demonstrate the system’s capabilities, we aimed to show that OPM-MEG hyperscanning can be used to measure brain activity whilst two players hit a table-tennis ball back and forth to one another. Unlike the guided touch task, where we expected temporally smooth head movements, we expected this task to generate movements that were quicker and more unpredictable. Despite such movement (maximum translations from the starting position in any one trial were 50.0 mm and 64.8 mm for Participants 1 and 3, respectively, and the maximum rotations were 17.0° and 17.4°). We again expected to observe a decrease in beta power in the left motor cortices during the period, where the game was played relative to rest.

The experimental setup is shown in Figure 5a. Participant 1 again wore an individualised scanner-cast (Figure 2a), whilst Participant 3 (Figure 2c, male, aged 41, height 188 cm) wore a generic 3D-printed helmet (co-registration as above). The participants stood ~80 cm apart, each holding a table-tennis bat in their right hand and were instructed to hit a table-tennis ball back and forth to each other for 5 s, following an audio cue. A second audio cue instructed the participants to stop their rally and rest for 7 s. This was repeated 25 times. The movement of the two helmets was again tracked using the OptiTrack camera system throughout the experiment. Trials, where the ball was dropped, were noted and excluded from data analysis (two dropped balls). The remnant field over each helmet was again nulled using the matrix coil (Figure 3b, the field decreased from 3.8 ± 2.8 nT to 1.6 ± 1.3 nT (a factor of 2.4) for Participant 1 and from 5.7 ± 2.9 nT to 2.0 ± 1.2 nT (a factor of 2.9) for Participant 3, we note that decreased nulling performance is due to the larger participant separation meaning fewer unit coils were contributing to the nulling). An audio cue signalled the participants to begin playing the game, after 5 s a second cue signalled the participants to stop the rally and rest for 7 s. This process was repeated 25 times and movement of the helmets during the experiment was recorded. Data were processed using a beamformer to derive an image showing the spatial signature of beta modulation between task (2 s < t < 4 s) and control (10 s < t < 12 s) windows. A time–frequency spectrum was also extracted from the peak of the beamformer image.

### 4.2. Results

Figure 5b shows the beamformer images overlaid on an anatomical MRI, and the Ŧ values were thresholded to 70% of the maximum value. The spatial signature suggests activation in the motor cortex as expected. Figure 5c shows the time–frequency dynamics of oscillatory power, revealing a reduction in the amplitude of beta activity during the task.

In addition to overall beta modulation (i.e., the difference between playing the game and resting) we also expected that, following each strike of the ball, a small amplitude increase in beta power should occur [17]; assuming consistent timings, we expected this effect should alternate between participants (e.g., we expect a peak in activity for Participant 1, and a trough in Participant 3 just after Participant 1 has hit the ball). Analysis was performed to probe the presence of this relationship. Figure 5d shows beta envelopes from both participants; data in the 3 s to 6.5 s time window were extracted and are shown inset in Figure 5e (blue for Participant 1, red for Participant 3). The normalised, unbiased, autocorrelation and cross-correlation of the two timecourses were computed for a maximum lag of 3.5 s (i.e., the full duration of the data segment). Figure 5e shows that the beta envelopes evolve in anti-correlation, with a lag of ~0.6 s between participants. This observation of the correlation of the amplitude envelope of oscillatory brain activity in two participants carrying out a single task highlights the power of hyperscanning.

## 5. Discussion

Brain stimulation in functional imaging is often provided by artificially controlled events, which take place in restrictive environments. Whilst useful, such experiments are of limited utility for the understanding of how the human brain works in its native surroundings. Multi-modal stimuli, such as audio-visual footage and immersive (real or virtual) environments, are now routinely deployed to investigate brain function during spontaneous, interactive events which more closely mimic real life. Such naturalistic settings are crucial for collecting ecologically valid neuroscientific data; indeed it has been postulated that the evolution of the human brain is closely linked to a need for complex social interactions [56]. For this reason, the importance of developing neuroimaging platforms that can interrogate brain function in naturalistic settings is paramount. The necessary technology to image brain function has so far been lacking either in performance or viability. Our work shows the potential of OPM-MEG as a hyperscanning platform capable of direct detection of electrophysiological responses, with millisecond temporal and millimetre spatial precision, during natural, live interactions.

Here we utilised the matrix coil to compensate remnant magnetic fields over two separate volumes to facilitate hyperscanning. However, participant movement away from these volumes was deliberately limited in our experiments (i.e., participants’ feet remained planted throughout recordings). This is because large measured field changes (e.g., due to sensor movement through a non-zero-field) move OPMs away from their zero-field operating point, inducing non-linearities and cross-axis projection errors in the OPM response which cause sensor gain and reduce the accuracy of neuronal source modelling [39,57,58]. Our previous work [39] extended the coil control software to account for the low-frequency field changes induced by movements by imposing feedback controllers on the OPMs’ sensitive outputs during a recording and updating the coil calibration matrix via optical tracking of the helmet, and real-time calculation of the field produced by each unit coil over the OPM array as the sensors moved. This development enabled ambulatory participant movement during a recording, and an extension of the technique to two helmets will be required to enable the full potential of OPM-MEG hyperscanning.

Here we present simple demonstrations that only hint at the possibilities for OPM-MEG hyperscanning. Previous work has shown myriad possibilities exist: An excellent example is the study of interactions between babies and their parents—indeed, past studies have employed EEG hyperscanning to show how the brains of a mother and baby demonstrate oscillatory synchronisation during normal social interactions, and that features of social interaction (e.g., eye contact) modulate the level of synchronisation [10]. This prior work demonstrated the power of hyperscanning, but it was based upon technology that is limited: EEG is highly motion sensitive, spatial resolution is limited (particularly in infants, where electrical potentials are distorted by the fontanelle) and high frequencies (beta and gamma oscillations) are disrupted by artefact. OPM-MEG technology has the potential to overcome these limitations. Similarly, it offers possibilities for new clinical investigations, for example, of social interaction in disorders, such as autism.

## 6. Conclusions

We performed the first OPM-MEG hyperscanning experiments. Wearable MEG combined with matrix coil active field control enables naturalistic scanning scenarios allowing two people to interact. With further development, OPM-MEG could become a powerful platform for the investigation of social neuroscience.

## Figures and Tables

**Figure 1 sensors-23-05454-f001:**
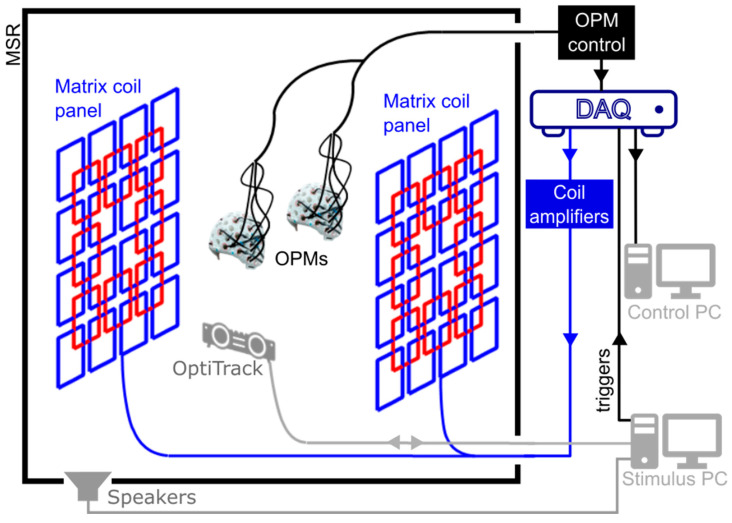
A schematic representation of the OPM-MEG system. The system is housed in a magnetically shielded room (MSR). OPMs are interfaced with a series of data acquisition devices. Data from the OPMs are used to drive the matrix coil field nulling process, before an MEG recording begins. Optical tracking of the helmets is performed to monitor motion during a session. Instruction is passed to the participants via auditory cues controlled using a separate stimulus PC.

**Figure 2 sensors-23-05454-f002:**
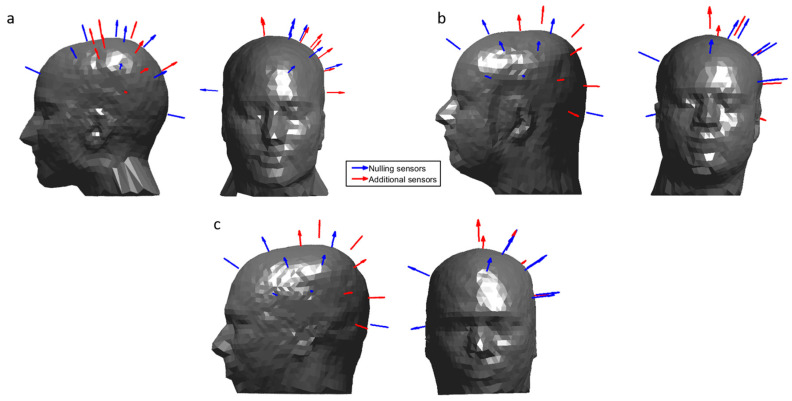
The position and orientation of the OPMs used in each experiment. Sensors that were used to inform the field nulling are shown in blue and additional sensors are shown in red. OPMs were concentrated over the left sensorimotor cortex, with additional sensors placed at the right and the front of the head, to inform the nulling process. (**a**) Sensor layout for Participant 1 during the two-person touch and ball game tasks. (**b**) Sensor layout for Participant 2 during the two-person touching task. (**c**) Sensor layout for Participant 3 during the two-person ball game.

**Figure 3 sensors-23-05454-f003:**
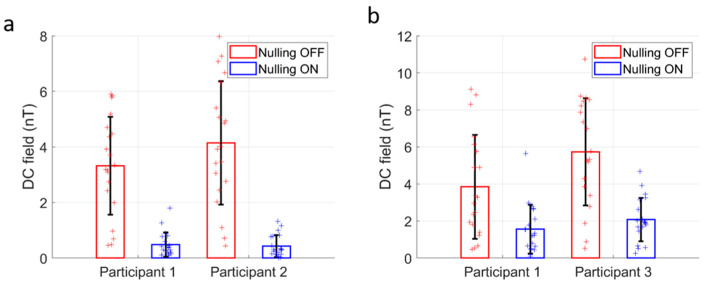
Effect of field nulling. (**a**) The strength of the DC field, reported by the 48 total field measurements from 24 OPMs (12 per participant), with and without the matrix coils active, during the 2-person touch task. The error bars show standard errors across sensors. (**b**) Equivalent to (**a**) for the 2-person ball game.

**Figure 4 sensors-23-05454-f004:**
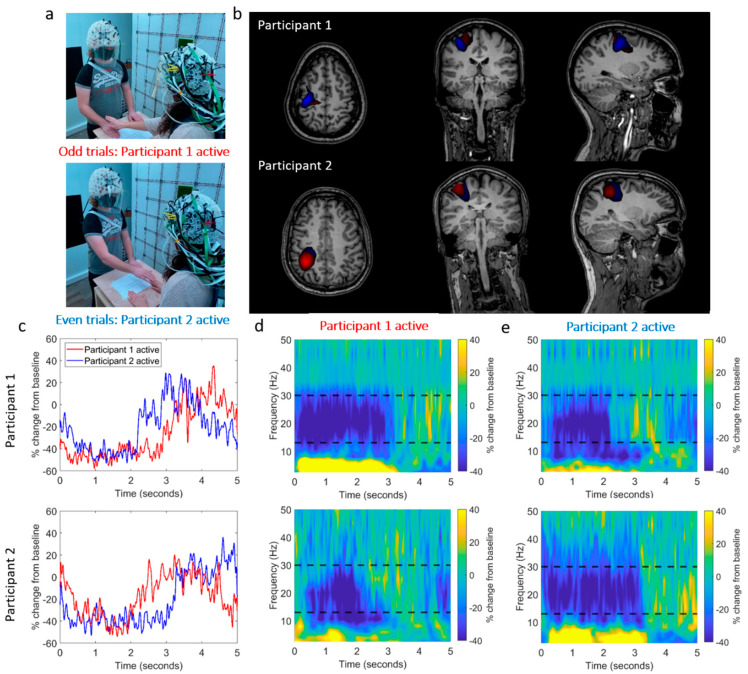
OPM-MEG data collected during a two-person naturalistic touching experiment. (**a**) Two participants stood on either side of a table. In odd-numbered trials, Participant 1 strokes the right hand of Participant 2, with their right hand. In even-numbered trials, the roles are reversed. (**b**) Beamformer images show the spatial signature of beta band modulation (thresholded to 80% of the maximum value). The odd trials (Participant 1 active) are shown in red and the even trials (Participant 2 active) are shown in blue. The spatial pattern suggests activity in the sensorimotor regions. (Note that there is a large overlap, so the blue overlay is partially obscured). (**c**) Timecourses showing the trial-averaged envelope of beta oscillations, extracted from the left sensorimotor cortex (peak in the beamformer images). Data from Participants 1 and 2 are shown in the top and bottom rows, respectively. In both cases, the red trace shows data recorded when Participant 1 was touching Participant 2. Similarly, (**d**,**e**) show time-frequency spectra of virtual electrodes at the peak of the beamformer images for each participant for both cases. Black dashed lines show the beta band.

**Figure 5 sensors-23-05454-f005:**
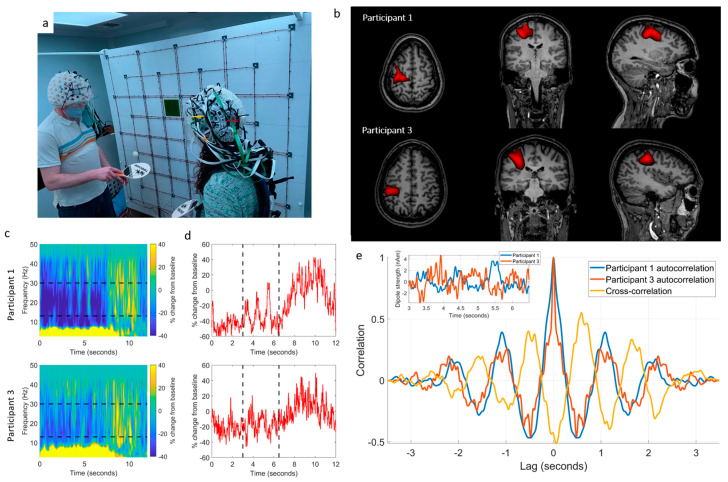
OPM-MEG data collected during a two-person ball game. (**a**) Two participants each held a table-tennis bat in their right hand and hit the ball back and forth to each other; a 5 s rally was followed by 7 s rest. (**b**) Beamformer images of beta band modulation between task and rest (thresholded to 70% of the maximum value). The spatial pattern suggests beta power reduction in the sensorimotor regions during the rally. (**c**) Time–frequency spectrograms were extracted from the left sensorimotor cortex for Participant 1 (top) and Participant 3 (bottom). Black dashed lines show the beta band. (**d**) Timecourses of the envelope of beta band activity (again Participant 1 top, and Participant 3 bottom). Data suggest anti-correlation between 3 and 6.5 s (marked with black dashed lines). (**e**) Inset: the timecourses extracted in the 3–6.5 s window and overlaid. Main: comparison of the autocorrelations of the two extracted timecourses (blue/red) with their cross-correlation (yellow) reveals anti-correlation with a lag of ~0.6 s between the participant’s brain activity. The photographs shown are of the authors.

## Data Availability

All data and code are available on request from the corresponding author.

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
