# Peer review of "Naturalistic Hyperscanning with Wearable Magnetoencephalography"

_sensors, 2023, doi:10.3390/s23125454_

Round 1
Reviewer 1 Report
You can move some informations from introduction to the discussion section. Congratulations for this work. Very innovator.
Reviewer 2 Report
In the submitted manuscript, the authors present an approach to perform OPM-MEG hyperscanning experiments based on their matrix coil system. In my opinion, all the methods and experiment results seem reasonable. I therefore support for publication of the manuscript, after addressing the comment written below.
(1) I noticed that there are some sort of hyperparameters in Methods section, such as the regulation parameter alpha and the gain coefficient G, that are empirically set to produce a satisfactory experiment result. It would be helpful if the authors could provide a more detailed explanation of how these hyperparameters were chosen, including why the regulation parameter was set to 1% of the maximum singular value of the matrix. Additionally, it would be useful to know what would happen if these hyperparameters were not correctly set.
(2) The authors mentioned that they performed field nulling using measurements of the amplitude of two field components from 10 OPMs. I am curious whether there would be a difference in field nulling performance if three field components were used instead of two, or if a single field component was used.
(3) I noticed that the total square side length of the 4x4 grid used in the authors' setup is almost the same as the distance between the two planes, which is approximately 150 cm. Typically, a pair of Helmholtz coils consists of two identical magnetic coils separated by a distance equal to the radius of the coil, which can minimize the nonuniformity of the field at the center of the coils. Is there a particular reason why the authors chose a distance that differs from the characteristic of Helmholtz coils?
(4) I came across a recently published paper in NeuroImage by the authors (doi: 10.1016/j.neuroimage.2023.120157), which provides a more detailed description of the matrix coil methods. I am interested to know if the matrix coil methods used in the submitted manuscript are the same as those described in the published paper.
(5) When recording the MCG signal of participants in motion, I am curious about how much artifacts would be induced, in addition to the gradient field factors, such as the magnetic fields produced by muscles and the unstable outputs of the OPMs due to motion acceleration. It would be beneficial if the authors could address these potential artifacts and their impact on the results of the experiments.

Reviewer 3 Report
The manuscript demonstrates the feasibility of hyperscanning using wearable magnetoencephalography (MEG) based on optically pumped magnetometers (OPMs) by conducting experimental studies on a few human subjects in two social, interactive tasks. Developing a matrix coil active control for field nulling to compensate for large and unpredictable subject motion their demonstration shows the effectiveness of the OPM-MEG approach in providing the direct neurophysiological features of cognitive tasks, high spatial and temporal resolutions, and robustness in naturalistic environments. The results show a few neurophysiological signatures of social interaction, specifically activation of the sensorimotor areas in the beamformer images and the modulation of beta band oscillation and their inter-subject correlation from the recorded MEG data.The rigor of analytical and experimental studies is powerful. The write-up is well-organized and clear. The study presents the feasibility and potential of a new technology which promises great improvements in studying neural correlates of brain cognitive functions. The scientific statements are relatively sufficiently supported by the presented results, however the small sample size and lack of discussion on relating the neurophysiological finding to the existing literature may weaken the scientific conclusions and therefore validation of the results.
Comments:
Line 260: the mean and standard deviation is over which variables? trials? sensors?
Line 275: are the odd and even trials randomly interleaved?
Line 278: what evidence or findings in the literature derive this hypothesis?
Line 311: what is the resolution of each band and overlapping steps? what was is bandwidth of the signals (bandpass filtered) going into the Hilbert transform? was there any specific reason for using Hilbert transform for calculating the frequency spectrum rather than wavelet or Fourier transforms?
Line 317: considering that the task window is 05s:2s and control window is 3s:4.5s, figure 4c shows that there is reduction in beta power during the task but enhancement during the control; is there any reasoning for the control effect?
Line 339: why is it expected to see a decrease in beta power? Figure 5d shows no difference in the beta modulation during the task 3s:6.5s, compared to the baseline (before 3s).
Line 352: it should be noted that the decrease is not statistically significant according to figure 3b; it could be because of small sample size or a non-optimal design.
Line 363: the text says 70% level for the threshold but figure 5 says 80%.
Line 367: again where does this expectation come from?
Line 377: the lag based on figure 5e looks much smaller than 0.6s? did you mean 0.06s?
Figure 4: legends for panel d and e are missing.
Figure 4c: the modulation effect difference between participant 1 active vs participant 2 active switches/reversed after 4s? is there any reasoning?
Figure 5d: it is not clear what this panel shows. How are participant 1 and 2 compared according to the plots?
